# Energy Absorption and Stiffness of Thin and Thick-Walled Closed-Cell 3D-Printed Structures Fabricated from a Hyperelastic Soft Polymer

**DOI:** 10.3390/ma15072441

**Published:** 2022-03-25

**Authors:** Ajeet Kumar, Luca Collini, Chiara Ursini, Jeng-Ywan Jeng

**Affiliations:** 1High-Speed 3D Printing Research Center, National Taiwan University of Science and Technology, Keelung Rd., Taipei 106, Taiwan; ajeetkumar@mail.ntust.edu.tw; 2Department of Mechanical Engineering, National Taiwan University of Science and Technology, Keelung Rd., Taipei 106, Taiwan; 3Department of Engineering and Architecture, University of Parma, Parco Area delle Scienze 181/A, 43124 Parma, Italy; luca.collini@unipr.it (L.C.); chiara.ursini@unipr.it (C.U.); 4Lunghwa University of Science and Technology, No. 300, Sec. 1, Wanshou Rd., Taoyuan 330, Taiwan

**Keywords:** additive manufacturing, cellular structure, support-less lattice structure, closed-cell lattice, hyperelastic material, 3D printing

## Abstract

This study analyses the energy absorption and stiffness behaviour of 3D-printed supportless, closed-cell lattice structures. The unit cell design is bioinspired by the sea urchin morphology having organism-level biomimicry. This gives rise to an open-cell lattice structure that can be used to produce two different closed-cell structures by closing the openings with thin or thick walls, respectively. In the design phase, the focus is placed on obtaining the same relative density with all structures. The present study demonstrates that closure of the open-cell lattice structure enhances the mechanical properties without affecting the functional requirements. Thermoplastic polyurethane (TPU) is used to produce the structures via additive manufacturing (AM) using fused filament fabrication (FFF). Uniaxial compression tests are performed to understand the mechanical and functional properties of the structures. Numerical models are developed adopting an advanced material model aimed at studying the hysteretic behaviour of the hyperelastic polymer. The study strengthens design principles for closed-cell lattice structures, highlighting the fact that a thin membrane is the best morphology to enhance structural properties. The results of this study can be generalised and easily applied to applications where functional requirements are of key importance, such as in the production of lightweight midsole shoes.

## 1. Introduction

### 1.1. Nature and Characterization of Lattice Structures

The word cell means a small, enclosed space or compartment. Cells of different morphology and size can cluster to form structures observed in nature. Wood, cork, sponge, and many other naturally occurring formations are all cellular structures. Such natural cellular structures can be found in the five ancient geometric solids called regular polyhedra or platonic solids [1,2]. ‘Lattice structure’ is defined as a cellular structure in which an interconnected network of surfaces repeats itself in a design space [3,4,5]. Hence, lattice structures are classified as a specific type of cellular structure. The mesoscale size of a unit cell, ranging from micrometres to one millimetre, allows lattices produced by Additive Manufacturing (AM) to be viewed both as a structure and a material; in other words, a “metamaterial”. Therefore, lattice structures can be considered materials in their own right. As is well known, an AM lattice structure exhibits its own set of material properties depending on the microstructural features of the representative unit cell [6]. All these cellular solids can be composed to have different shapes, forms, and tesselations.

Bhat et al. demonstrated the effect of tessellation on the mechanical and functional property, while the morphology of the unit lattice structure was the same [7]. The cellular structure based on topological space can be classified as two-dimensional (2D) or 3-dimensional (3D). Two-dimensional cellular solids are the simplest of all cellular solids, made of a two-dimensional array of polygons that fill a design space and are extruded in the third dimension. Such structures, for example, honeycomb lattices, have anisotropic properties and can also be termed 2.5D structures [8,9]. Three-dimensional cellular solids are instead polyhedral cells packed in three dimensions and can have either isotropic or anisotropic mechanical properties [3]. Lattice structures can be designed in two ways, (a) Truss-based [10] or (b) Plate-based [11]. By varying unit cell design parameters such as topology (connectivity), geometry (truss/plate dimensions), and morphology, the functional and mechanical properties of a lattice can be significantly affected, which would not be impossible with the bulk constituent materials [12].

The most important structural characteristics of lattice structures are the relative density, $\rho^{*}$, and the relative stiffness, *E^*^*, described as:

The relative density:(1)$$\rho^{*}=\frac{\rho_{l}}{\rho_{s}}$$
where *ρ_l_* and *ρ_s_* are the densities of the lattice and solid, respectively, which can be expressed in terms of the volume reduction coefficient (VRC) ϕ, ranging from zero to one, where zero indicates that the lattice is fully solid:(2)$$\phi =1-\frac{V_{L}}{V_{S}}$$
and (ii) the relative stiffness *E*^*^:(3)$$E^{*}=\frac{E}{\rho^{*}}$$
where *E* denotes Young’s modulus of the lattice. The mechanical properties of the cellular structure can be tuned by modifying optimisation parameters such as the thickness or diameter of the truss/surface element or size of the unit cell. Finally, to fulfil design or functional requirements, the topologies of open-cell structures are important to consider. A closed-cell lattice is usually designed by understanding the functional requirement of an open-cell lattice, which can be transformed into a closed-cell structure by closing the openings with thin or thick walls such that the functional requirements remain unchanged. In this case, two important design principles arise for closed-cell lattice structures:I.open-cells are closed with a relatively thin solid wall/membraneII.open-cells are closed with relatively thick solid wall/membrane [3,5,13]

These two principles can be employed to produce both local and global closed-cell lattice structures [14,15]. Local closed-cell lattice structures have fully closed unit cells that are tessellated in the design space. In contrast, global closed-cell lattice structures have open unit cells tessellated in the design space but enclosed from the outside.

### 1.2. Thermoplastic Polyurethane for Additive Manufacturing

Additive manufacturing considered sustainable technology could mitigate the effect of future climate change with distributed manufacturing and lattice structure that could help build parts lightweight with high speed [16,17]. The impact of additive manufacturing and its material development is seen in various fields, such as photopolymerization of the polymer under visible light [18,19]. Additive manufacturing of thermoplastic polyurethane (TPU) material is increasingly used in industrial fields, such as footwear, automotive, and aerospace, owing to its biodegradable nature, abrasion resistance, flexibility, and energy absorption capability [13,14]. Printed TPU does not exhibit intuitive behaviour due to its hygroscopic, viscoelastic and hyperelastic nature [20]. In order to understand the complex behaviour of this material, finite element (FE) modelling has been employed in parallel with experimental investigations. Designed structures are produced by AM via a material extrusion process (MEX) with a TPU filament of known nominal mechanical properties. Three-dimensional printing of this hyperelastic polymer with an open-source MEX process is challenging due to the viscosity and low Young’s modulus of TPU. Common problems in printing with flexible/soft filaments are frequent, including buckling and kinking, slippage of the filament in the feed module, clogging of the nozzle and absorption of moisture from the atmosphere. This study has minimised above problem by two ways:IDesign of support-less continuous surface-based lattice structure. This type of design can eliminate the requirement of filament retraction during printing. Repeated retraction of filament is normally seen during printing truss-based lattice structures as these are non-continuous surfaces.IIPrinting process parameter optimization. The retraction is not allowed as this is a major problem during soft-material printing. Repeated retraction is responsible for buckling, kinking, and filament slippage in the feed module. The hygroscopic nature of TPU is minimised by keeping the filament in the dry box during the 3D printing process.

At the same time, numerical analysis and matching the hysteresis and energy absorption of printed TPU is also very difficult due to variability of the resulting properties and anisotropy introduced by the layering nature of the process [21,22].

### 1.3. Analysis of Cellular Structures

The selected open and closed-cell lattice structures considered in the present work, bioinspired by the sea urchin (SU), have excellent functional and mechanical properties and can be successfully fabricated without support with the MEX process [15,16]. Recently, many researchers have used bioinspiration to design lattice structures for energy absorption applications [23]. Ha et al. designed a novel bioinspired honeycomb structure for energy absorption for lightweight panels [24]. The SU exhibits remarkable mechanical properties driven mainly by its structural morphology, exhibiting relevant functional and mechanical properties such as energy absorption and stiffness. Further, by designing the lattice structures for support-free printing, requirements for support structures for primary or secondary material within the lattice are eliminated, resulting in a considerable advantage in terms of cost and production time [25]. This can be performed with the MEX process, requiring no essential post-print operations to remove support structures from closed cells.

This study wants to analyse the relationship between specific design features and functional and mechanical properties of this kind of cellular lattices made in TPU. Relevant industrial applications are detectable, as the development of products based on filled or unfilled closed-cells. Where functional requirements are key considerations, such as personal protection gears, customized shoe midsoles, and other lightweight, high-performance panel or layer components.

## 2. Materials and Methods

### 2.1. Design of Closed Lattice Cell

The unit cell is the basic building block of a lattice structure. The topology of a unit cell characterises each lattice structure in determining its functional properties. The first step in the design phase is to select an imaginary rectangular prism or voxel of dimensions X × Y × Z, which acts as a bounding box for the unit cell. Values of X, Y, and Z may vary based on design specifications. Inside this bounding box, the plate/shell is connected or intersected to form the morphology of the unit cell according to the design requirements.

The design phases of the unit cell inspired by SU morphology having organism level biomimicry are illustrated in Figure 1. Based on this geometry, a local closed-cell lattice structure was developed. The lattice structure was supportless based on generating a primitive surface patch in Matlab 2021b©. These surface geometries were exported in Creo Parametric© environment for converting them to solid by adding and trimming material in the outer direction for the final local closed-cell design, as shown in Figure 1 [14,26].

The boundary equations defining the surface patch are described in Equations (4)–(9).
(4)$$\mathrm{Curve}\;\;1:\;Z\;=\;0\;\mathrm{plane}:\;\;{\left(x-a\right)}^{2}+{\left(y-a\right)}^{2}={\left(a/2\right)}^{2}$$
(5)$$\mathrm{Curve}\;\;2:\;X=a\;\mathrm{plane}:\;\;y^{2}+z^{2}={\left(a/2\right)}^{2}$$
(6)$$\mathrm{Curve}\;\;3:\;Y=0\;\mathrm{plane}:\;\;{\left(z-a\right)}^{2}+{\left(x-a\right)}^{2}={\left(a/2\right)}^{2}$$
(7)$$\mathrm{Curve}\;\;4:\;Z=a\;\mathrm{plane}:\;\;x^{2}+y^{2}={\left(a/2\right)}^{2}$$
(8)$$\mathrm{Curve}\;\;5:\;X=0\;\mathrm{plane}:\;\;{\left(y-a\right)}^{2}+{\left(z-a\right)}^{2}={\left(a/2\right)}^{2}$$
(9)$$\mathrm{Curve}\;\;6:\;Y=a\;\mathrm{plane}:\;\;z^{2}+x^{2}={\left(a/2\right)}^{2}$$

The openings shown in Figure 1e were then closed with thin and thick walls to understand the effect of their presence on stiffness and energy absorption, representing the main mechanical and functional properties of such structures. The thin wall was designed with a lower thickness than that of the open-cell shell. The thick wall was instead designed with a greater thickness than that of the open-cell shell, as illustrated in Figure 2 and Table 1. The third step in design is the tessellation of the unit cell with the concept of periodic tessellation having a unary connection of edge-to-edge type. Each unit cell shared a complete edge (open or closed) with their adjacent lattice for uniform load transfer. In this way, there was no bridge between adjacent unit cells [17], allowing supportless printing of lattice structure with the MEX process, as shown in Figure 2.

The final step comprised of completing tessellation of the lattice structure over the entire design space, which was chosen to be a cylindrical specimen. This procedure was very challenging due to overhangs of the lattice truss/plates near the boundary of design space, leading to failure of the printing process. The behaviour of both open and closed structures was then characterised and directly compared with experimental and numerical methods.

### 2.2. Additive Manufacturing

The designed lattice structures were optimised using the “design for additive manufacturing and post-processing” (DfAM&PP) concept for the MEX/FFF process (Flashforge dreamer©, Jinhua, China). DFAM&PP focuses on four crucial design parameters for successful printing of supportless structures, including minimum feature size, minimum thickness, minimum overhang angle, and minimum parallel ledges [27,28,29] (see Figure 3). These parameters differ according to the material, process, and 3D printer used for manufacturing.

The optimised MEX process for defect-free printing of TPU was characterised by a minimum feature size of 6 × 6 × 6 mm (Figure 3a), a minimum thickness of 0.2 mm (Figure 3b), and a minimum overhang angle of 50° (Figure 3c). As shown in Figure 3, no parallel ledges or bridges were printed, thus requiring no continuous support during fabrication. For each type of lattice structure, including open-cell and thin and thick-walled closed-cell structures, three specimens were printed with the parameters listed in Table 2. No special filament extruder was employed for the fabrication of the parts in TPU.

Figure 4 shows all the manufactured parts, with the red arrow representing the printing direction in all cases. Particular attention was paid to the ambient temperature, which was maintained between 22 and 25 °C. No post-process was performed on the printed lattice structures.

### 2.3. SEM Measurements

Defects such as sagging, distortion, improper layer adhesion, and porosity can change the mechanical and functional properties of 3D-printed lattice structures. The presence of such defects in the printed sample, both in the longitudinal and lateral directions, was observed using a JOEL JSM-6390LV, Japan, scanning electron microscopy (SEM). Each specimen was subject to titanium dioxide sputtering in both exposed surface directions to make the polymer surface conductive.

## 3. Experiments and Simulation

### 3.1. Experiments

A standard monotonic compression test at a constant strain rate is usually employed for the mechanical characterisation of MEX-fabricated structures. With this relatively simple test, the load-deformation and equivalent stress–strain behaviour can be determined and compared up to the densification regime. Owing to the hysteretic viscous behaviour of TPU, the load should be applied cyclically at least 10–20 times to evaluate the energy absorption of this material [30,31]. The stabilised cycle must then be analysed based on the area under the load curve in the hysteresis plot.

In this work, uniaxial compression tests were performed at a 5 mm/min strain rate using an MTS 104 pneumatic electromechanical material testing system equipped with a 10 kN load cell. Compression was performed at three levels, including 10, 20, and 30% of the sample height, *h*_0_, perpendicular to the printing layer direction. Load-displacement curves were obtained, from which the classical stress–strain relationship and energy absorption, *W_c,_* were calculated according to the following equations [3]:(10)$$\sigma_{N,c}=\frac{P_{c}}{A_{0,eq}}$$
(11)$$A_{0,eq}=\frac{V_{L}}{h_{0}}=\left(1-\phi \right)h_{0}^{2}$$
(12)$$\varepsilon_{N,c}=\frac{u_{c}}{h_{0}}$$
(13)$$W_{c}=\int_{\varepsilon =0}^{\varepsilon =\varepsilon_{0.4}}\sigma_{N,c}\varepsilon_{N,c}d\varepsilon$$
where *P_c_* and *σ_N,c_* are the compressive loads and nominal stress, *A*_0,*eq*_ is the equivalent cross-sectional area of the lattice, *u_c_* and *ε_N,c_* are the displacement and nominal compressive strain, and *W_c_* is the energy absorbed per unit volume. The stiffness of the specimen was calculated by interpolating the slope line over the load portion. The energy loss area was calculated after 20 loading cycles with the stabilised curve.

### 3.2. Simulation

The simulation approach was based on the analysis of a unit cell for each different type of 3D-printed lattice structure to characterise the behaviour of the entire structure, as shown in Figure 5. With this approach, the unit cell was a representative volume element of the full structure. Hence, it could be repeated without any limit over three dimensions to constitute a lattice structure. With this approach, correct boundary conditions, mechanical properties, and deformation behaviour of cells can be easily and quickly investigated, saving computational time. Different geometric configurations can be analysed and compared without needing to produce and test them all experimentally. Optimisation problems are also solvable with reasonable calculation resources. Morphological features are typically given as variable parameters in optimisation runs aiming to maximise a structure’s performance in terms of given properties such as the specific stiffness. In this work, two variable parameters were chosen for analysis: the wall and shell thicknesses, $T_{w}$ and $T=R_{2}-R_{1}$, respectively.

The Abaqus/CAE© 2022 (6.20) software suite was used for numerical simulation. The models were developed by importing CAE generated solid models into the software. The meshing of the solid cells was performed by an automatic algorithm based on the inner growth option. Meshing was performed on a test model adopting a convergence criterion of 1% deviation in stiffness. Linear solid tetrahedral elements with a hybrid formulation and a characteristic dimension of 0.8 mm were used, resulting in 1111–1441 nodes and 4189–5847 elements depending on the geometry.

Simulations were performed using the large deformation option in the explicit solver within the software. Movement of a rigid analytical surface at the top of the cell was imposed, with a small sliding option at the surface’s contact. Contact was governed by a friction coefficient of f=0.1, as results had been found to remain largely invariable for values between 0.05 and 0.3. Periodic boundary conditions were applied to the unit cell faces to fully reproduce the actual deformation behaviour. Specifically, periodicity was applied to the displacement of nodes on the cell-free faces with the ABAQUS© equation option. The studied FE unit cell models are shown in Figure 5.

The numerical results yielded ‘*RF_i_*’ as the reaction force at *N_C_* due to the compressive displacement of constrained nodes in the *z*-direction, from which the load curve and cell stiffness, *K*_0_, were obtained according to the following relations:(14)$$F_{T}=\overset{N_{C,z}}{\underset{i=1}{\sum }}RF_{z,i}$$
(15)$$F\left(z\right)=K_{0}z+q\;\;for\;z\in \left[0,{\left.z\right|}_{F_{max}}\right]$$

The total reaction force, *F_T_* = *f*(*z*), was interpolated with a linear function over the compression strain range up to *F_max_*, from which the stiffness was calculated using Equation (15). The stiffness of a lattice structure with certain dimensions can be estimated in a simple manner. From the data obtained for the unit cell, an equivalent, homogeneous virtual material was defined, characterised by stiffness based on the resulting section-to-length ratio. Finally, by applying 20 cycles of compression release, the numerical hysteresis loop was plotted. This was performed to ensure that steady-state hysteresis was reached, which was generally the case after 15 cycles. For the purposes of analysis, data relating to the twenty-first cycle were taken into consideration. Energy absorption was determined as the integral of the load-displacement graph, as shown in Figure 6. The percentage coefficient of absorbed energy was calculated as:(16)$$\Omega_{c}=\frac{W_{1}}{W_{2}}\left(\%\right)$$
where $W_{1}$ and $W_{2}$ are the absorbed and total energy, respectively. The total energy was defined as the sum of the absorbed and released energy during the 21st cycle of the unit cell. During the experimental phase, the behaviour of each cell was analysed at 20% and 30% deformation.

### 3.3. Material Model

TPU is a complex material exhibiting hyper-elastic and viscoelastic properties [20]. In addition, fused filament fabrication (FFF) produces an anisotropic, layered material structure that is usually no more than half as strong in the tangential direction than it is in the transversal direction [32]. By comparing a tensile test of the ASTM D638 Type 4 dog-bone sample with the different strain energy potential models offered by Abaqus©, it turned out that the second-order Ogden model better reproduced the experimental tensile curve in the stretch of deformation of interest, as shown in Figure 7. Following the Ogden formulation, the N-order polynomial strain energy potential U*^def^* is expressed as follows:(17)$$U^{def}=\overset{N}{\underset{i=1}{\sum }}\frac{2\mu_{i}}{\alpha_{i}^{2}}\left({\bar{\lambda }}_{1}^{\alpha_{i}}+{\bar{\lambda }}_{2}^{\alpha_{i}}+{\bar{\lambda }}_{3}^{\alpha_{i}}-3\right)+\overset{N}{\underset{i=1}{\sum }}\frac{1}{D_{i}}{\left(J_{el}-1\right)}^{2i}$$
where ${\bar{\lambda }}_{i}$ is the deviatoric stretch, *J_el_* is the elastic volume ratio and *α_i_*, *μ_i_*, and *D_i_* are material model parameters reported in Table 3. Hysteretic behaviour, responsible for energy dissipation under repeated cyclic loading, is modelled within ABAQUS©. The mechanical response is made up of an equilibrium part following stress relaxation after a long time (network *A*) and a non-linear, time-dependent function that produces a perturbation from the equilibrium state (network *B*). The model considers the total stress as the sum of the network *A* and network *B* stresses. In mathematical terms, the effective creep strain rate in network *B* (${\dot{\epsilon }}_{B}^{cr}$) is given by Equation (18):(18)$${\dot{\epsilon }}_{B}^{cr}=A{\left[\lambda_{B}^{cr}-1+E\right]}^{C}{\left(\sigma_{B}\right)}^{m}$$
where $\lambda_{B}^{cr}-1$ and *σ_B_* are the nominal creep strain and effective stress in network *B*, respectively. Five parameters fully define the material model. These include the stress scaling factor (*S*), which is the ratio of the network *B* and network A stresses under instantaneous loading, the exponent *m*, which is generally greater than one and expresses the network *B* stress dependence of the steady-state creep strain rate, the exponent *C* [−1, 0], which characterises the network *B* creep strain dependence of the creep strain rate, and the constants *A* and *E*, which characterise the effective creep strain rate. The network *B* stress scaling factor and creep parameters, together with an optimisation procedure for FE modelling of cylindrical specimens subject to compression loading, were selected based on the literature [20]. The chosen parameters are indicated in Table 4.

The employed hysteresis model allowed not only strain-dependent hysteresis loops during unloading to be reproduced but also the permanent deformation after each compression-release cycle. This made it possible to determine the energy absorption per cycle, as shown previously.

## 4. Results

### 4.1. SEM Analysis

SEM images of the fabricated open-cell and thin and thick-walled closed-cell lattice structures are illustrated in Figure 8 in both the longitudinal (LD) and transversal (TD) directions. The printed open-cell lattice topology in the LD (Figure 8a) did not exhibit imperfections such as sagging, distortion, layer peeling or porosity. A few microscopic pores between the layers were instead present in the TD, as seen in Figure 8d. After careful analysis, it was found that the presence of these pores, due to air bubbles or under-extrusion, did not adversely affect the structural performance of the TPU-printed open-cell structures.

Longitudinal pores in the print direction were observed on the surface in both the LD and TD for the thin-walled closed-cell lattice structure (Figure 8b). In the TD, microscopic pores between the two layers of the outer and inner walls were present, as can be seen in Figure 8e. These microscopic pores were observed in a linear pattern up to the end of the thin wall. This effect was due to the under-extrusion of the material from the nozzle during the printing of this thin layer. Unlike the thin-walled structures, no longitudinal pores or layer peeling in the print direction were observed in the thick-walled structures, as seen in Figure 8c. In the TD, continuous microscopic pores were present in the inner and outer layers of the thick-walled lattice structure, as can be observed in Figure 8f. This was due to the small gap left during the connection of the infill to the boundary wall layer. However, these microscopic pores did not significantly affect the thick-walled closed-cell structural performance [15].

### 4.2. Functional Response of Structures

Figure 9 shows images of the specimens tested at the three considered levels of compression. As can be observed, 30% compression led to a very large degree of deformation with barrelling and densification of the specimens. The obtained hysteresis curves are shown in Figure 10 for the various tested levels of deformation, while the full set of results is shown in Table 5. Based on these outcomes, it can be observed that the mechanical and functional properties of the open-cell and thin-walled closed-cell structures exhibited comparable behaviour in terms of stiffness and energy absorption.

In contrast, thick-walled closed-cell structures exhibited a 20% reduction in performance. In general, the stiffness of the considered hyperelastic polymer material appeared to decrease with increasing deformation. At low strain levels, all tested samples exhibited high levels of resilience; however, as deformation increased, the stiffness decreased.

In relation to the FE simulations, unit cells with applied periodic boundaries (see Figure 11) exhibited very similar behaviour to the experimental outcomes both in terms of stiffness and energy absorption, even if a clear deviation was evident at low strain levels. The deformation behaviour, i.e., logarithmic strain in the compression direction of open-cell (Figure 11a), thin wall (Figure 11b) shows barrelling whereas thick wall (Figure 11c) negative barrelling effect is seen.

A plot comparing numerical and experimental data is shown in Figure 12, where it is clearly visible how the open and thin-walled closed-cell structures exhibited a similar mechanical response in terms of stiffness and energy absorption. On the other hand, results differed for the thick-walled closed-cell structures.

## 5. Discussion

### 5.1. Mechanical Properties

As discussed previously, the stiffness of the printed TPU lattice structures decreased as the degree of compression increased, with little variation in initial response between each cell type. It is important to note that the open-cell structure had no walls, with material concentrated at the ribs. As the cell ribs were free to bend more easily than the walls during compression tests, they could better withstand the load. Therefore, the open-cell structure possessed material where it was most useful, with the ribs providing the most geometrically efficient topology. This consideration, however, was no longer valid for high strain levels where they caused significant deformation at the beginning of the densification regime.

The results of FE simulations were in agreement with experimental outcomes for the open-cell structures, while the stiffness of the closed-cell structures was overestimated. It is likely that the thick-walled structures shrunk in the transversal direction, which will be discussed in more detail below.

Experimental values of the energy absorption coefficient, Ω*_c_*, exhibited low sensitivity to the level of deformation, ranging from 20–25% for 10–30% compression. These values were in accordance with previous findings by the same authors for similar relative densities [25]. The numerical approach using FE software considered lower dispersion of the deformation energy at low levels of compression, giving rise to evident differences in the experimental results. The hysteretic material model was certainly responsible for this effect as it was calibrated for a given deformation range and did not consider dependence on other parameters such as the temperature. Moreover, although strain rate effects were considered within the numerical model, it is likely that they require further dedicated testing. Finally, since results were obtained by homogenisation of the response of a single cell, the numerical models with boundary conditions failed to reproduce the real contact state between closed-cell walls, which may have drastically reduced the deformation. In relation to this aspect, Figure 13 displays an analysis of the samples’ simulated and experimental radial dilatation. From this analysis, it is possible to determine that the numerical simulation:-was accurate for the closed thick-walled structures;-overrated the thin-walled structures by a factor of 1.25;-underestimated the open-cell structures by a factor of 1.5.

The observed behaviour, which was in contrast with the stiffness prediction, did not significantly change with variations in friction coefficient over the range considered within the investigation (*f* = 0.05–0.3).

**Figure 13 materials-15-02441-f013:**
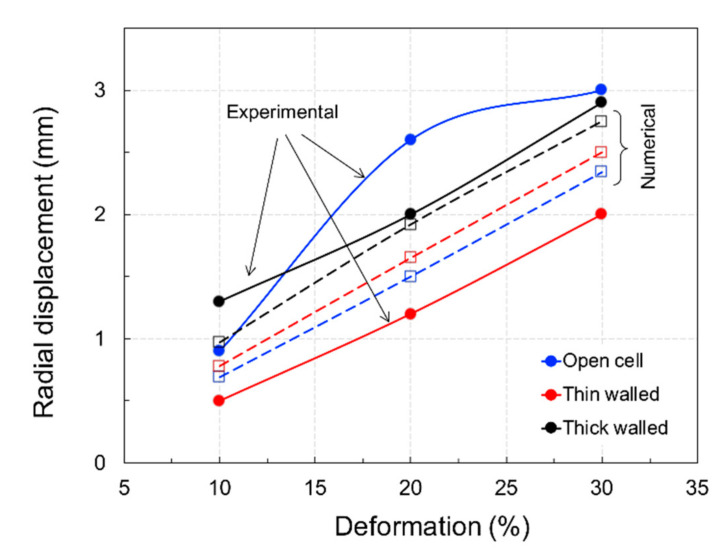
Experimental and numerical radial dilatation of specimens.

As discussed in detail below, this effect may have been due to anisotropy between material layers, with elastic constants being different between lattices. However, the study shows that it is possible to compare the behaviour of different lattice structures via FE modelling prior to conducting experiments. Despite the sophistication of the material model, numerical analysis can be an effective virtual design tool to study the maximum variation in load response of unit cells for different geometries and identify and optimise performance.

### 5.2. Effect of AM Process

Specimens obtained with the MEX process were generally non-uniform at different levels, as observed during SEM analysis. Anisotropy in elasticity and strength was also readily observed. More broadly, numerous process parameters that adversely affect the mechanical properties of the final printed polymeric object were highlighted and are summarised below:

Pores: additive manufacturing of TPU filament with MEX often left pores between layers, which was due to the hygroscopic nature of TPU, under-extrusion of material while printing thin sections or retraction of the filament into the nozzle during printing [26,27,28], as can be seen in Figure 8.

Building orientation: some compression tests performed on the MEX-printed samples exhibited different values of failure load depending on how the load was applied with respect to the deposition plane, with horizontal orientation leading to a higher strength than vertical orientation [33,34,35]. Building orientation had a significant effect on the difference between FE results and experimental outcomes.

Temperature: the difference in temperature between adjoining layers during solidification of the filament caused residual stress, shrinkage and distortion of the fabricated part. The temperature difference between the build chamber and filament, therefore, had a significant effect [36,37].

These effects appeared to be amplified in the thin-walled structure, which was characterised by the minimum printing thickness. A higher concentration of micro-voids occurred in layers during printing, as seen in Figure 8b,e, drastically reducing the ability of this structure to withstand loads [15,38]. Lattice structures obtained with the MEX process were, therefore, already very sensitive to process parameters used during the 3D printing phase. In relation to the compressive mechanical response of polymeric samples obtained via the MEX process, a comprehensive review of the literature over the last two decades, starting from 1996 [38], revealed that the process is influenced by the raster orientation (transverse, axial), air bubbles or gaps, layer width, model temperature, layer thickness, building orientation, raster angle, raster width, and infill (low, high, double dense, solid) [39]. Therefore, it is possible to state that the combined effect of these parameters, the material, processing conditions, and environment greatly influence the final response of the printed product. This is of considerable importance for appropriate analysis of the numerical results by considering the combined effect of all possible conditions that alter the mechanical response of printed samples.

### 5.3. Effect of Hyperelasticity, Suggestions Relating to Design

The compressive behaviour of the open and thin-walled closed lattice structures was similar in terms of mechanical and functional properties; however, the performance metrics of the thick-walled results were 20% lower. This was in contrast with previous results obtained by the same authors for closed-cell lattice structures printed with PLA filaments, where the best response in terms of stiffness and energy absorption was obtained with a thin-walled structure compared to other geometries [14].

For lattice structures printed in TPU, a thermoplastic elastomer [20,40], the stiffness was only slightly affected by the shell thickness. The open-cell structure was 1.2 mm thick, the shell thickness of the thin-walled closed structure was 1.0 mm (0.6 mm for the walls), and the shell thickness of the thick-walled closed structure was 0.66 mm (1.2 mm for the walls). Since the flexural stiffness *D* of a shell is proportional to its thickness *T*,
(19)$$D\propto E\cdot T^{3}$$

A variation ± of 70% was expected between the lattices if the walls do not bend, or a variation of 40% was expected if the walls do bend. None of these extreme bounds was met. Because of the extended strain range for TPU, which is an elastomer, the compression of such structures can extend to very high levels. Thus, the bending of cell ribs and the closing of walls became strongly non-linear both geometrically and in terms of the material response.

It is possible that the ribs and walls unevenly supported bending, as the material was distributed to achieve constant density. Figure 14 displays the strain in the vertical direction (load direction), *ε*_22_*,* for the ribs and walls at 10–20% to 30% compression levels. It is evident that the ribs were only subject to tensile loading for the open and thin-walled cells at the extrados. As the wall thickness increased, bending decreased both in the ribs and walls, as shown in Figure 11. Moreover, the open-cell structure was more efficient in bending. If the compressive strain is subtracted and the thickness considered, an equivalent non-dimensional bending moment *M_b_*^*^ can be calculated considering the strain gradient across the sections, yielding the results plotted in Figure 15. Here, the individual contribution of the ribs and walls (when present) can be appreciated, showing that walls increasingly supported bending, unloading the shell, as their thickness increased. However, bending in the two closed-cell geometries was always less than that occurring in the open-cell structure. This can explain the observed difference between the predicted and observed stiffness, supporting the hypothesis that the deposition process had a strong influence on the mechanical properties, accentuated at large thickness.

## 6. Conclusions

The properties of lattice structures with open-cells and thin and thick-walled closed-cell geometries printed in TPU via MEX have been investigated. Wall thickness, useful for designing closed unit cells from open topologies, was analysed in terms of its effect on the functional and mechanical properties. Extensive laboratory testing and sophisticated numerical modelling were performed to investigate the determinant design features for optimising structural performance. The mechanical and functional properties of the lattices were found to exhibit unintuitive behaviour, from which new design considerations could be drawn:-AM of TPU allows closed-cell lattice structures to be designed that mimic the sea urchin and behave as non-linear springs with excellent energy absorption properties;-closed cells with thin walls are more effective for optimising mechanical and functional-properties compared to those with thick walls. Hence, a closed-cell lattice structure should be designed with a thin wall to maintain or enhance an open-cell lattice structure’s mechanical and functional requirements;-FE simulations can be employed to reveal manufacturing defects, including the fact that printed TPU samples are anisotropic, with the degree of anisotropy depending on the level of deformation;-strong interaction between the layering process and structural properties must be considered during the design phase of lattice structures produced via MEX.

## Figures and Tables

**Figure 1 materials-15-02441-f001:**
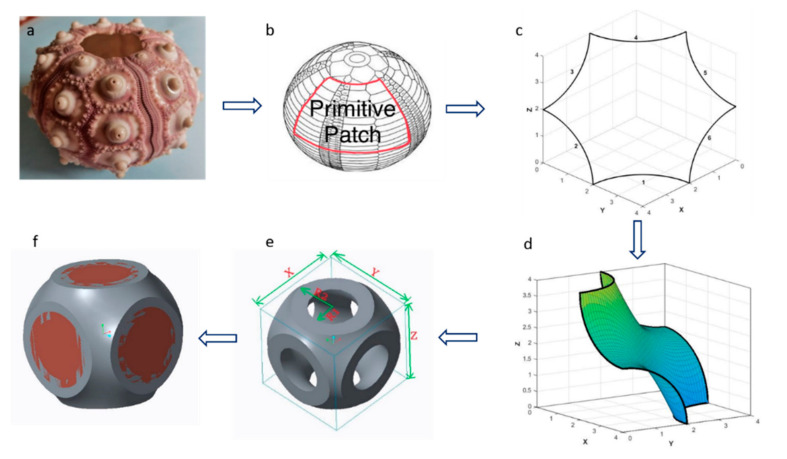
The biomimetic local closed-cell lattice structure based on SU morphology (**a**) SU morphology (**b**) primitive patch (**c**) primitive patch developed by boundary equation (**d**) surface generation on the boundary equation and mirroring of this surface patch for final design in Creo (**e**) final design of supportless open-cell lattice structure in Creo parametric (**f**) closed-cell designed by closing the openings of the open-cell lattice structure.

**Figure 2 materials-15-02441-f002:**
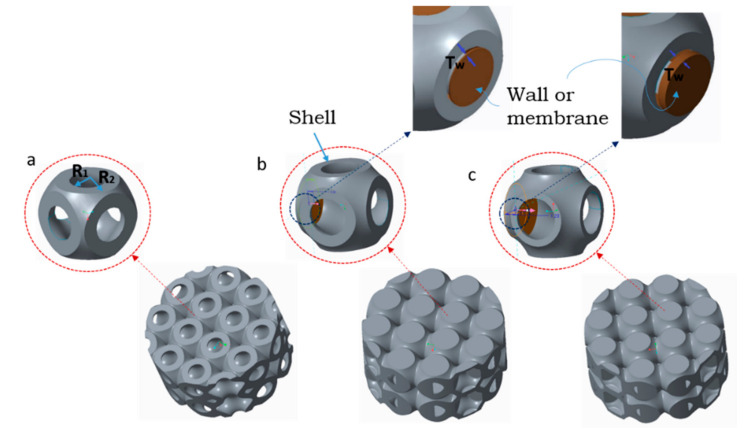
Designed model (**a**) open-cell lattice (**b**) thin-walled closed-cell lattice (**c**) thick-walled closed-cell lattice.

**Figure 3 materials-15-02441-f003:**
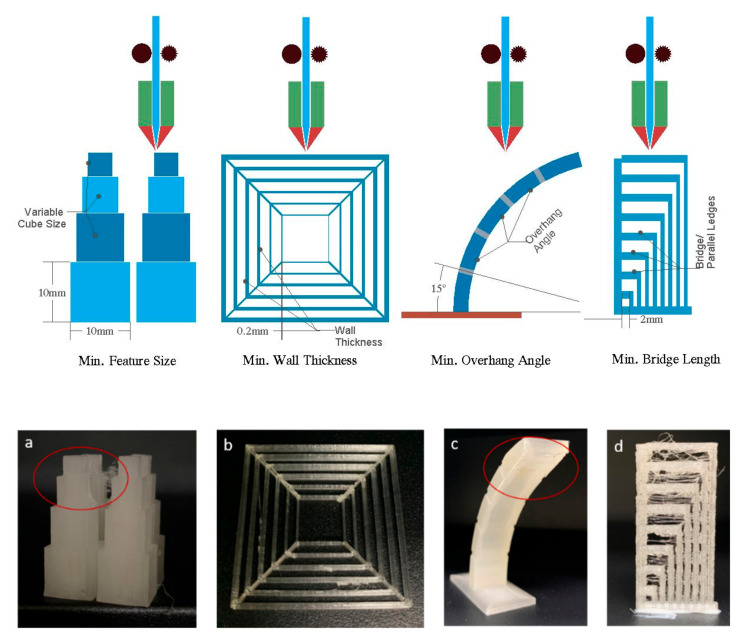
Four parameters considered within design for additive manufacturing and post-processing (**a**) minimum feature size (**b**) minimum wall thickness (**c**) minimum overhang angle (**d**) minimum bridge length.

**Figure 4 materials-15-02441-f004:**
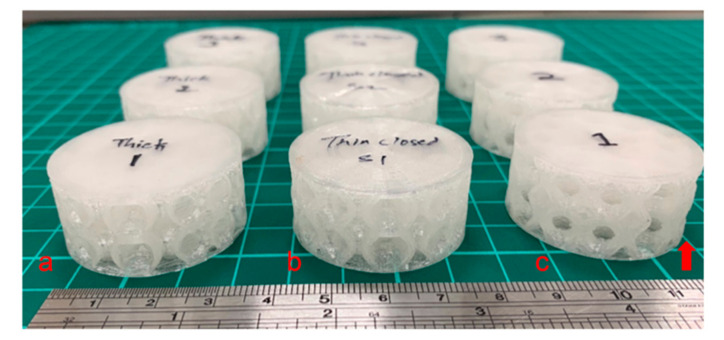
AM using MEX process: (**a**) thick-wall closed-cells; (**b**) thin-wall closed-cells; (**c**) open-cells. The arrow represents the build direction.

**Figure 5 materials-15-02441-f005:**
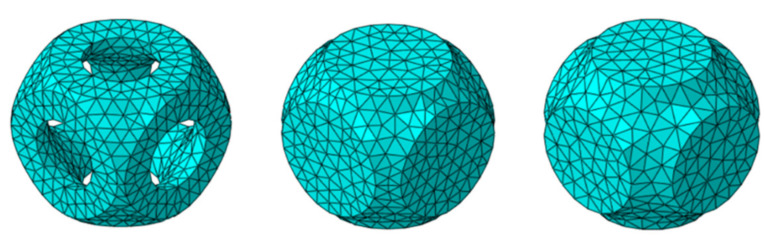
FE models of open, thin- and thick-walled closed unit cells.

**Figure 6 materials-15-02441-f006:**
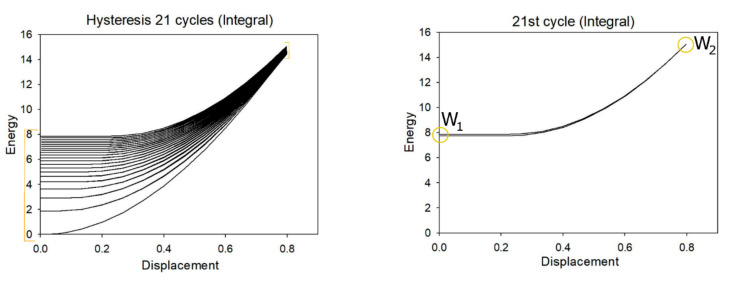
Schematic of numerical calculation of absorbed energy.

**Figure 7 materials-15-02441-f007:**
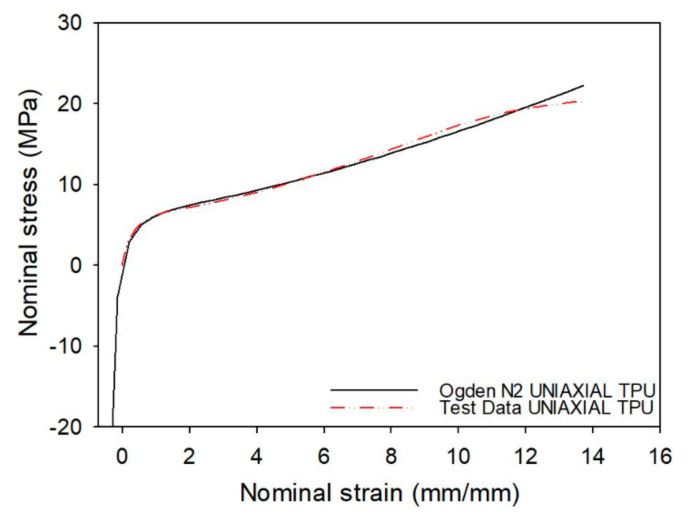
Experimental and Ogden model (N = 2) behaviour for the employed TPU.

**Figure 8 materials-15-02441-f008:**
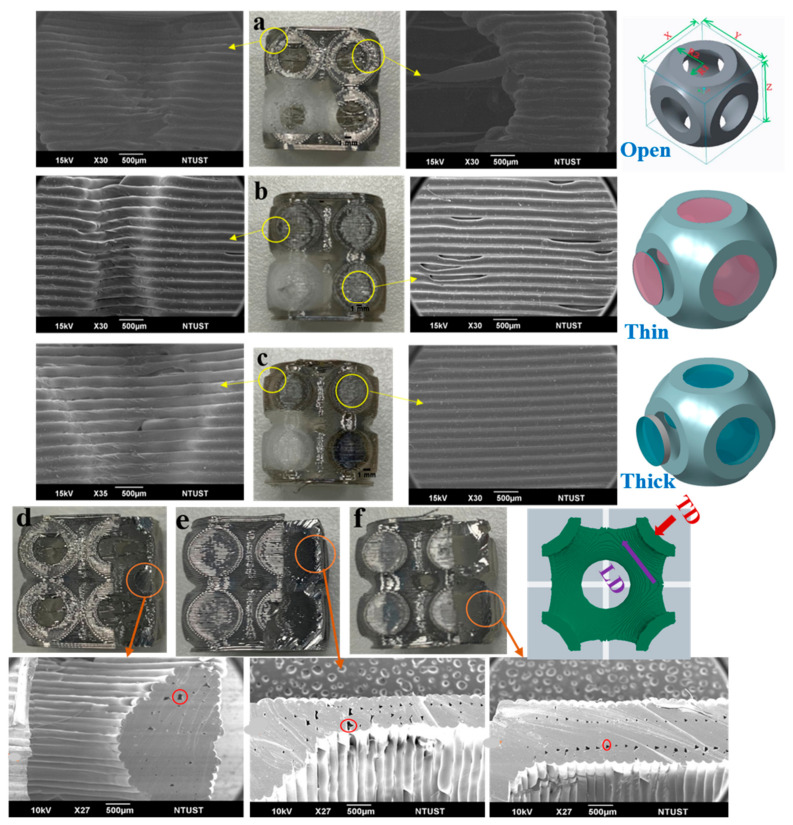
SEM images of the TPU layer deposition in both longitudinal direction (LD) and transversal directions (TD) (**a**) open-cell LD direction no defects are observed (**b**) thin-wall LD under-extrusion layer deposition is observed (**c**) thick-wall LD no defects are observed (**d**) in open-cell TD random micro-pores encircled with red colour is observed (**e**) continuous micro-pore encircled in red colour are seen in thin-wall TD (**f**) In thick-wall TD too micro-pores are observed.

**Figure 9 materials-15-02441-f009:**
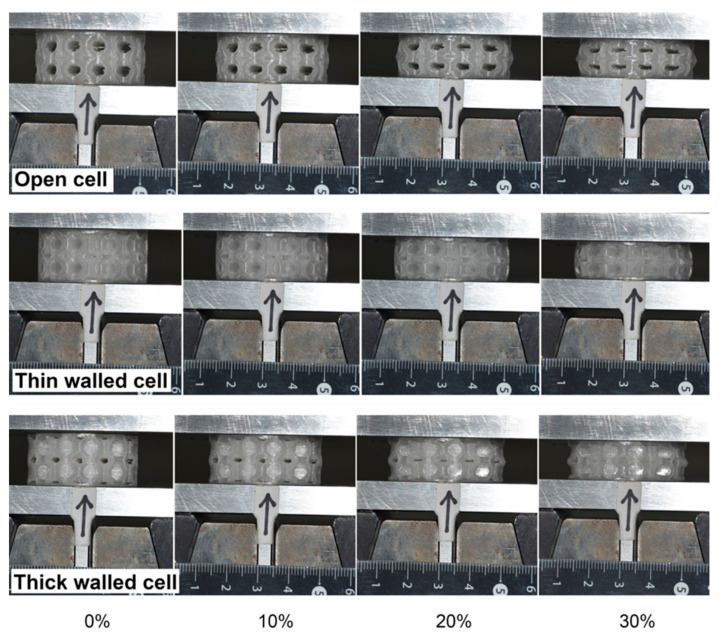
Images obtained during compression testing of all printed lattice structures.

**Figure 10 materials-15-02441-f010:**
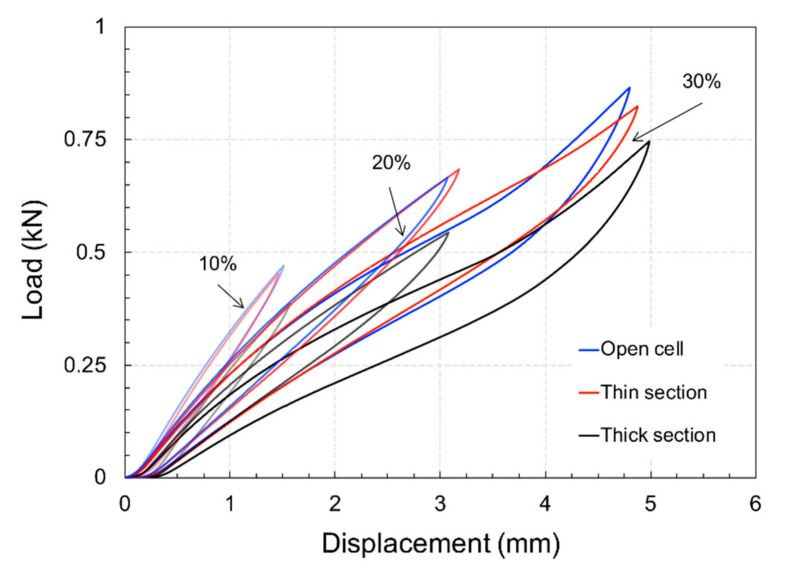
Experimental load-displacement curves.

**Figure 11 materials-15-02441-f011:**
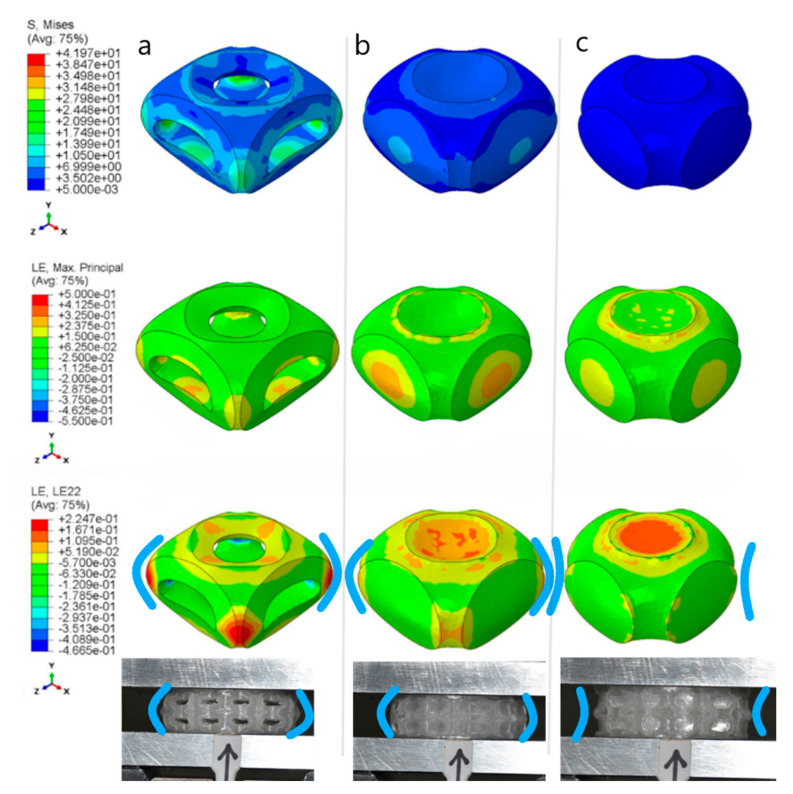
Unit cells 30% deformed: Von Mises equivalent stress, Principal and compressive strains ε_22_. The deformation behaviour of open (**a**), thin (**b**) shows barrelling, whereas thick wall (**c**) negative barrelling effect is seen.

**Figure 12 materials-15-02441-f012:**
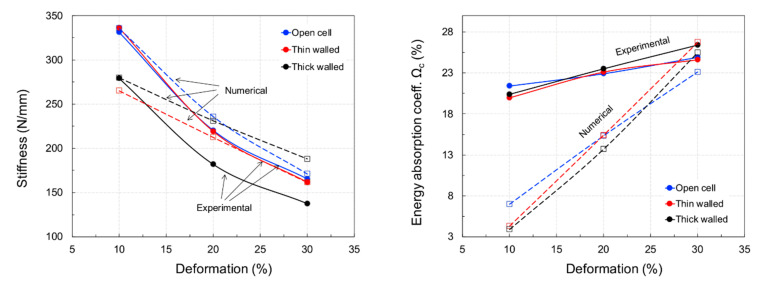
Experimental and numerical results.

**Figure 14 materials-15-02441-f014:**
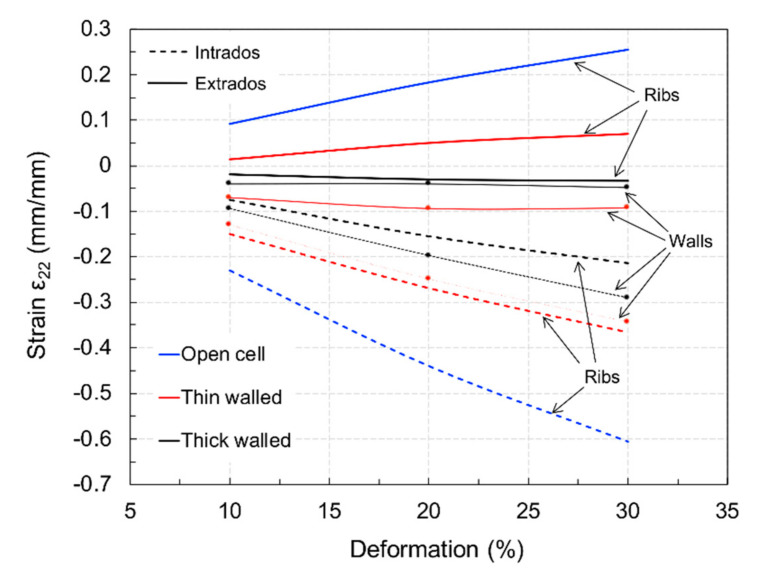
Strain in ribs and walls at the intrados and extrados of the cell structures.

**Figure 15 materials-15-02441-f015:**
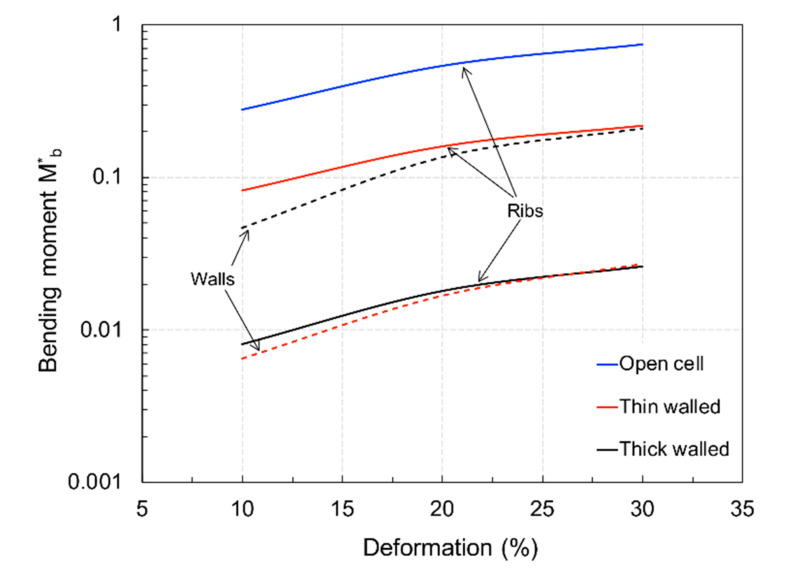
Comparison of bending load in the ribs and walls of the structures.

**Table 1 materials-15-02441-t001:** Parameters of analysed lattice structures.

Cell Structure	Material	Specimen/ Unit Cell Size (mm)	Open Shell Thickness (T) (mm)	Closing Wall Thickness (T_w_) (mm)	VRC ϕ (%)
Open-cell	TPU	Φ37 × 16/8 × 8 × 8	1.2	–	56.9
Thin-walled			1.0	0.6	57.0
Thick-walled			0.66	1.2	56.6

**Table 2 materials-15-02441-t002:** MEX parameters used for printing all lattice structure and dog-bone samples with TPU filament.

Extruder Diameter (mm)	Extruder Temperature (°C)	Platform Temperature (°C)	Layer Thickness (mm)	Infill (%)	Print Speed (mm/min)
0.4	230	70	0.2	100	1100

**Table 3 materials-15-02441-t003:** Ogden material model parameters for the employed TPU.

Order	*μ_i_*	*α_i_*	*D_i_*
1	0.4055	2.4580	6.1616 × 10^−3^
2	6.1298	−1.9004	0.0000

**Table 4 materials-15-02441-t004:** Hysteresis parameters for the employed TPU.

*S*	*A* (s^−1^MPa^−m^)	*m*	*C*	*E*
2.2	0.556	4.0	0.0	0.01

**Table 5 materials-15-02441-t005:** Experimental and simulated lattice structure stiffness and energy absorption.

Lattice Structure	*K*_0_ (N/mm)						W*c* (J/m^3^)						Ω*c* (%)					
*ε_N_*_,_*_c_* (%)	10		20		30		10		20		30		10		20		30	
	Exp	FEA	Exp	FEA	Exp	FEA	Exp	FEA	Exp	FEA	Exp	FEA	Exp	FEA	Exp	FEA	Exp	FEA
Open-cell	331.0	335.3	220.3	235.8	165.5	171.0	7.58	1.21	27.73	8.32	59.61	25.12	21.4	7.0	22.9	15.3	24.9	23.1
Thin-wall	336.0	264.6	219.2	212.3	161.7	161.4	6.84	1.08	32.72	13.33	61.52	48.23	20.0	4.3	23.1	15.4	24.6	26.7
Thick-wall	278.9	279.7	181.9	231.1	137.5	188.1	6.35	0.98	25.04	12.06	57.11	49.13	20.4	3.9	23.5	13.7	26.4	25.5

## Data Availability

The data presented in this study are available on request from the corresponding author.

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
