# Peer review of "Energy Absorption and Stiffness of Thin and Thick-Walled Closed-Cell 3D-Printed Structures Fabricated from a Hyperelastic Soft Polymer"

_materials, 2022, doi:10.3390/ma15072441_

Round 1

Reviewer 1 Report

Thank you for the interesting paper. I have no suggestions for improvement

Author Response

Reviewer 1

Overall, this paper is well-organized, but the author wants to address some questions to make this paper clearer to a general audience.

  1. Numeric labeling of each section is confusing. For example, section 1 is introduction, and the next section is materials and methods. However, the labeling for both sections are 1.

Thanks for pointing this mistake. This happened due to conversion in MDPI format. We have corrected this problem.

  1. The author mentioned common problems in printing with flexible filaments including buckling, slippage, etc, can the author briefly explain how such factors/issues were minimized/prevented to reduce effects on mechanical/functional properties.

This study has minimised above problem by two ways;

  1. design of support-less continuous surface-based lattice structure. This type of design can eliminate the requirement of filament retraction during printing. Repeated retraction of filament is normally seen during printing truss-based lattice structures as these are non-continuous surfaces.

  1. printing process parameter optimization. The retraction is not allowed as this is a major problem during soft-material printing. Repeated retraction is responsible for buckling, kinking and slippage of filament in the feed module. The hygroscopic nature of TPU is minimised by keeping the filament in dry box during 3D printing process.

Same has been included in the manuscript section 1.2 and highlighted.

  1. For SEM tests,

3-1. Schematics of design model are highly recommended to added to Figure 8, near the real prints. The printing, longitudinal (LD) and transversal (TD) directions need to be marked on the models to enable audience to understand the test better.

Thanks for the suggestion, now we have improved the figure for better understading.

3-2. Does the author have any proof, which can be test results or references to draw the conclusion that the presence of pores did not adversely affect the structural performance of the TPU-printed open cell or thick-walled closed-cell structures. Can the author explain what is “under-extrusion”

The typical defects are observed in FFF process during printing soft material like TPU. These effects appeared to be amplified in the thin-walled structure, which was characterised by the minimum printing thickness. A higher concentration of micro-voids occurred in layers during printing, as seen in Figure 8(b,e), drastically reducing the ability of this structure to withstand loads. Lattice structures obtained with the MEX process were therefore already very sensitive to process parameters used during the 3D printing phase. Similar defects were also observed by Prajapati et. al. which has been no cited.

Under-extrusion- During printing the thin wall the the filament oozing from FFF nozzle tends to reduce. This under-oozing leads to underextrusion leaving gap between the layer. Same cannot be observed during printing the thick wall as oozing of filament from nozzle is 100%.

  1. The author needs to give extra explanation in the caption of Figure 11. What does the color indicate?

This indicate logarithmic strain in direction 2 (compression). This has also been added in the manuscript section and highlighted.

  1. Is the trend that closed cells with thin walls exhibit higher stiffness and energy absorption than those with thick walls universal for this series of lattice structures or material-properties dependent. In simulation works, did mechanical properties such as modulus of TPUs be taken into consideration?

This is universal for the all the lattice structure closed cell. The mechanical and functional behavior of unit lattice structure at higher relative density (>0.2) is highly depended on the morphology i.e structure. At lower relative density the mechanical and functional property is highly depended on the material. When closing the open cell with thick wall the morphology changes and main mechanical property is derived from the wall and not from open cell morphology. This effect would be highly seen in case of elastic plastic material like ABS or PETG.

Yes, TPU is modeled as hyper elastic material via specific model which consider real stress-strain relationship starting from the nominal tensile curve.

Reviewer 2 Report

Overall, this paper is well-organized, but the author wants to address some questions to make this paper clearer to a general audience.

  1. Numeric labeling of each section is confusing. For example, section 1 is introduction, and the next section is materials and methods. However, the labeling for both sections are 1.
  2. The author mentioned common problems in printing with flexible filaments including buckling, slippage, etc, can the author briefly explain how such factors/issues were minimized/prevented to reduce effects on mechanical/functional properties.
  3. For SEM tests,

3-1. Schematics of design model are highly recommended to added to Figure 8, near the real prints. The printing, longitudinal (LD) and transversal (TD) directions need to be marked on the models to enable audience to understand the test better.

3-2. Does the author have any proof, which can be test results or references to draw the conclusion that the presence of pores did not adversely affect the structural performance of the TPU-printed open cell or thick-walled closed-cell structures. Can the author explain what is “under-extrusion”.

  1. The author needs to give extra explanation in the caption of Figure 11. What does the color indicate?
  2. Is the trend that closed cells with thin walls exhibit higher stiffness and energy absorption than those with thick walls universal for this series of lattice structures or material-properties dependent. In simulation works, did mechanical properties such as modulus of TPUs be taken into consideration?

Author Response

This paper presents the investigation on the energy absorption and stiffness behaviour of 3D printed supportless, closed-cell lattice structures. It is an interesting paper. However, there are some concerns that should be addressed before accepting

  1. Please check the format such as section number,

Thanks for pointing this mistake. This happened due to conversion in MDPI format. We have corrected this problem.

  1. In Fig. 7, the author used the dog bone to determine the mechanical properties. Did the author consider the printing direction?
  2.  

Yes, we have printed the dogbone sample in tensile pulling direction thereby reducing or eliminationg the effect of layer. The print direction and pull direction is same.

  1. Please specify clearly why the friction coefficient of ? = 0.1 was used.

We tried different values for f in the simulations, from 0.04 to 0.2, and no significant changes occured in the response: hence f=0.1 is chosen based on literature data about metal-plastic conatct in dynamic, lubricated condition https://www.tribology-abc.com/abc/cof.htm.

  1. Please compare deformation shape between simulation and experiment.

Figure 11 has been replaced to compare the experimental and simulation deformation shape.

  1. There are many bio-inspired structures that have been developed in the recent year such as

https://doi.org/10.1007/s10853-018-3163-x

https://doi.org/10.1016/j.compositesb.2019.107496

It should be discussed in the introduction part

We have included in the introduction section 1.3 of this manuscript and highlighted.

Reviewer 3 Report

This paper presents the investigation on the energy absorption and stiffness behaviour of 3D printed supportless, closed-cell lattice structures. It is an interesting paper. However, there are some concerns that should be addressed before accepting

  1. Please check the format such as section number,
  2. In Fig. 7, the author used the dog bone to determine the mechanical properties. Did the author consider the printing direction?
  3. Please specify clearly why the friction coefficient of ? = 0.1 was used
  4. Please compare deformation shape between simulation and experiment.
  5. There are many bio-inspired structures that have been developed in the recent year such as

https://doi.org/10.1007/s10853-018-3163-x

https://doi.org/10.1016/j.compositesb.2019.107496

It should be discussed in the introduction part

Author Response

Reviewer 3

Jeng and coworkers performed a study about the absorption and stiffness behavior of 3D printed supportless, closed-cell lattice structures. The experiments are well done and support their conclusion. The work itself is also novel for both academic and industrial study. It is an interesting work to publish on Materials, but the following comments must be addressed.

  1. The number before every subtitle is a mess. Every subtitle number in the paper starts from 1, 1.1, 1.2, etc. Please correct them. The current version is really hard to follow.

Thanks for pointing this mistake. This happened due to conversion in MDPI format. We have corrected this problem.

  1. There are some typos and grammar mistakes across the paper. Please correct them.

We have checked it and improved the grammatical mistake.

  1. A couple of corrections on SEM are needed. The image quality of 8a is low. The authors should adjust the focus and brightness when taking the images. I suggest the authors redo the SEM for Figure 8a if it is possible. Besides, a note of what is the red circle in Figure 8e-f should be added in the figure captions. In figure 8b, the scale bar on the photo is covered by the yellow circle and should be better organized.

We have changed the figure 8 and mentioned about the figure details in the figure caption.

  1. The introduction part needs to be revised and expanded. I strongly suggest the authors introduce additive manufacturing/3D printing as an individual paragraph, as 3d printing is the main method for this work. Some recent advances in polymer materials printing and different printing methods such as photo 3D printing, inject printing and others could be introduced, and some recent related works should be cited (https://doi.org/10.1039/D1PY00705J; https://doi.org/10.1016/j.apmt.2020.100804).

In the manuscript, the citation is included in the introduction section 1.2.

Reviewer 4 Report

Jeng and coworkers performed a study about the absorption and stiffness behavior of 3D printed supportless, closed-cell lattice structures. The experiments are well done and support their conclusion. The work itself is also novel for both academic and industrial study. It is an interesting work to publish on Materials, but the following comments must be addressed.

  1. The number before every subtitle is a mess. Every subtitle number in the paper starts from 1, 1.1, 1.2, etc. Please correct them. The current version is really hard to follow.
  2. There are some typos and grammar mistakes across the paper. Please correct them.
  3. A couple of corrections on SEM are needed. The image quality of 8a is low. The authors should adjust the focus and brightness when taking the images. I suggest the authors redo the SEM for Figure 8a if it is possible. Besides, a note of what is the red circle in Figure 8e-f should be added in the figure captions. In figure 8b, the scale bar on the photo is covered by the yellow circle and should be better organized.
  4. The introduction part needs to be revised and expanded. I strongly suggest the authors introduce additive manufacturing/3D printing as an individual paragraph, as 3d printing is the main method for this work. Some recent advances in polymer materials printing and different printing methods such as photo 3D printing, inject printing and others could be introduced, and some recent related works should be cited (https://doi.org/10.1039/D1PY00705J; https://doi.org/10.1016/j.apmt.2020.100804).

Author Response

(The authors gave the same response as above.)

Round 2

Reviewer 4 Report

Current version is very good and reads well to me.